# Genetic Control of Efficient Nitrogen Use for High Yield and Grain Protein Concentration in Wheat: A Review

**DOI:** 10.3390/plants11040492

**Published:** 2022-02-11

**Authors:** Wan Teng, Xue He, Yiping Tong

**Affiliations:** 1The State Key Laboratory for Plant Cell and Chromosome Engineering, Institute of Genetics and Developmental Biology, Chinese Academy of Sciences, Beijing 100101, China; tengwan@genetics.ac.cn (W.T.); hexue@genetics.ac.cn (X.H.); 2College of Advanced Agricultural Sciences, University of Chinese Academy of Sciences, Beijing 100049, China; 3The Innovative Academy of Seed Design, Chinese Academy of Sciences, Beijing 100101, China

**Keywords:** *Triticum aestivum*, nitrogen use efficiency, grain yield, grain protein concentration, root morphology, nitrate transporter, nitrogen assimilation

## Abstract

The increasing global population and the negative effects of nitrogen (N) fertilizers on the environment challenge wheat breeding to maximize yield potential and grain protein concentration (GPC) in an economically and environmentally friendly manner. Understanding the molecular mechanisms for the response of yield components to N availability and assimilates allocation to grains provides the opportunity to increase wheat yield and GPC simultaneously. This review summarized quantitative trait loci/genes which can increase spikes and grain number by enhancing N uptake and assimilation at relative early growth stage, and 1000-grain weight and GPC by increasing post-anthesis N uptake and N allocation to grains.

## 1. Introduction

Wheat (*Triticum aestivum*) is the most widely cultivated cereal in the world, providing 20% of the calories and protein in the global human diet [1]. As the world population is predicted to be over 9 billion by 2050, the demand for wheat will increase by 60% compared to 2010 [2]. Nitrogen (N) is an essential nutrient, and often the most critical yield-limiting factor in crop production [3]. Among the crops, wheat received the highest amount of N fertilizers, with 18.2% of global use, followed by maize with 17.8% and rice with 15.2% [4]. A global N budget assessment in cereal production systems for 50 years (1961 to 2010) revealed that approximately 44.6% of N harvested by wheat crops was derived from applied fertilizer-N, indicating most of the applied N was not used by wheat crops [5]. Fertilizer N not recovered by crops can cause environmental problems of nitrate pollution of waters and the pollution of the atmosphere with nitrous oxide, other oxides of N, and ammonia [6]. The Wheat Initiative Strategic Research Agenda suggests increasing N use of over 60% of the applied amount to increase wheat production by at least 60% by 2050 [2]. Therefore, it is required to increase wheat yield in an economically and environmentally friendly manner. Such as breeding wheat with improved N use efficiency (NUE) and designing variety-specific recommendations of N fertilizer management [2,7,8,9].

The genetic control of N use-related traits and strategies in improving NUE in wheat have been reviewed by several authors [8,10,11]. As selecting yield and grain protein concentration (GPC) both are important in wheat breeding, and the improvement of both traits depends on efficient N use, this review will summarize N uptake and assimilation quantitative trait loci (QTL)/genes which coordinate the formation of yield and yield components and N allocation to grains.

## 2. Definition of N Use-Related Traits

NUE and its components N uptake efficiency (NUpE) and utilization efficiency (NUtE) are defined as below [12].
(1)$$\mathrm{NUE}=\frac{\mathrm{GY}}{\mathrm{Ns}}$$
(2)$$\mathrm{NUpE}=\frac{\mathrm{Gt}}{\mathrm{Ns}}$$
(3)$$\mathrm{NUtE}=\frac{\mathrm{GY}}{\mathrm{Nt}}$$
where GY is grain yield per unit area, Nt is total N in the plant (grain + stove) at maturity per unit area, and Ns is N supply or rate of fertilizer N per unit area. GY, Nt, and Ns are all expressed in the same units.

N harvest index (NHI) can be expressed as:
(4)$$\mathrm{NHI}=\frac{\mathrm{GY}\times \mathrm{GNC}}{\mathrm{Nt}}$$
where GNC is grain N concentration. According to the above four equations, we can get the following equations:
(5)NUE=NUpE×NUtE
(6)$$\mathrm{GY}=\mathrm{NUtE}\times \mathrm{Nt}=\frac{\mathrm{GY}}{\mathrm{Nt}}\times \mathrm{Nt}$$
(7)$$\frac{\mathrm{NUtE}}{\mathrm{NHI}}=\frac{\mathrm{GY}}{\mathrm{GY}\times \mathrm{GNC}}=\frac{1}{\mathrm{GNC}}\;\mathrm{or}\;\mathrm{NUtE}=\frac{\mathrm{NHI}}{\mathrm{GNC}}$$

## 3. Genetic Gain for N Use- and Yield-Related Traits

Understanding historical trends in N use-related traits during wheat breeding is important to design strategy in improving the NUE of future varieties while maximizing yield potential and GPC [13]. Although wheat breeding was in most cases not targeted to improve NUE [7], significant genetic gains have been observed for N use-related traits around the world (Table 1). The rates of genetic gain for NUpE, NUtE, N harvest index, and post-anthesis N uptake were comparable in wheat main production regions, including China, Europe, and the United States (Table 1). The four studies listed in Table 1 all observed a much lower rate of genetic gain for NUpE and NUtE than for grain yield, and a lower rate for N harvest index than for harvest index. These data indicate that NUpE, NUtE, and N harvest index have great potential to be improved in future wheat breeding. Considering that GNC is negatively correlated with NUtE (Equation (7)) and GPC (GPC = 5.7 × GNC) is a key element of wheat end-use value, NUtE should be improved by increasing N harvest index (N remobilization), but not by reducing GNC or GPC. As such, scientists should explore genes that can increase NUpE and more efficiently allocate the absorbed N to grains to simultaneously improve both yield and GPC.

## 4. Associations of N Uptake and Assimilation Genes with Meta-QTL for N Use- and Yield-Related Traits

Nitrate is the main N resource for wheat. Nitrate uptake by roots is mediated by the low-affinity transport system encoded by NRT1/NPF family genes and the high-affinity transport system encoded by NRT2 family genes. Once nitrate enters plant cells, it is reduced to nitrite by nitrate reductase (NR) in the cytosol, and nitrite is then translocated to the plastids and reduced to ammonium by the nitrite reductase (NiR). Ammonium can be assimilated into glutamine (Gln) and glutamate (Glu) through the glutamine synthetase (GS)/glutamate synthase (GOGAT) cycle [17].

The wheat genome contains 331 *NRT1/NPF* and 46 *NRT2* genes [18,19]; however, only a few of these transporters have been functionally characterized. The two *NPF* genes (TraesCS1B02G038700 and TraesCS1D02G214200) and one *NRT2* gene (TraesCS6A02G030800) displayed nitrate transport activity when expressed in *Xenopus* oocytes [18]. TraesCS1D02G214200 (*TaNRT1.1B-1D2*) is closely linked with a meta-QTL for yield-related traits (Table 2) and is an orthologue of rice *OsNRT1.1B*. The elite allele of *OsNRT1.1B* can improve NUE and grain yield of rice (*Oryza sativa*) [20]. TaNRT2.5-3B required a partner protein TaNAR2.1 to give nitrate transport activity in oocytes and mediated long-distance transportation of nitrate from roots to grains [21]. *TaNRT2.1-6B* encoded a dual-affinity nitrate transporter and was associated with N harvest index and grain yield. Overexpressing *TaNRT2.1-6B* increased grain yield and N uptake under low- and high-N conditions in a pot experiment [22]. Genotyping of a panel of 254 diverse wheat cultivars and landraces by the wheat 660k Axiom Array indicated that wheat breeding did not affect the polymorphisms of most of the *NPF* and *NRT2* genes [18]. When comparing the physical positions of the *NRT1/NPF* and *NRT2* genes with the locations of meta-QTL for N use- and yield-related traits [11,23,24], we found that several *NPF* and *NRT2* genes are located in the interval of the meta-QTL (Table 2), providing cues for exploring the contribution of N transporters to the natural variation in N use- and yield-related traits.

Analysis of Chinese Spring reference sequence (http://plants.ensembl.org/Triticum_aestivum/Info/Index, accessed on 15 December 2021) revealed that the genes encoding NR, NiR, GS, and GOGAT all present in a small gene family in wheat, with three *TaNR*s, one *TaNiR*, three cytosolic *TaGS1*s, one plastic *Ta**GS2,* and two *GOGAT*s in each of the wheat sub-genomes. As a small number of primary N-assimilation genes mediate the assimilation of the inorganic N absorbed by wheat plants into amino acids, they should have a critical role in determining NUE in wheat. When comparing the physical positions of these primary N-assimilation genes with the locations of meta-QTL for N use- and yield-related traits [11,23,24], we found that 15 of these 30 N-assimilation genes locate in the interval of the meta-QTL (Table 2). Cross-genome meta-QTL analysis also revealed the crucial role of N-assimilation genes in mediating NUE in crops including wheat [25]. Haplotype variations of *TaGS2-2A*, -*2B* and *-2D* were associated with root system architecture-, N use- and yield-related traits [26], and the elite allele of *TaGS2-2Ab* has been successfully used to engineer wheat with improved NUE and yield under both high- and low-N conditions [27]. *TaNADH-GOGAT-3B* has been suggested as a candidate contributing to the natural variation of NUE in wheat [25], more recently overexpression of *TaNADH-GOGAT-3B* has been shown to increase N uptake and yield in wheat [28]. In rice, allelic variation at *OsNR2* contributed to the difference in nitrate assimilation capacity and NUE between the *indica* and *japonica* rice subspecies, with *indica OsNR2* exhibiting greater NR activity [29]. The elite allele of *OsNiR1* with increased NiR activity significantly improved rice regeneration [30], and increasing *OsNiR1* expression enhanced NUE by promoting spike number in rice [31]. In maize (*Zea mays*), the *GS1* genes *ZmGln1.3* and *1.4* linked with QTL for NUE-related traits and were specifically involved in the control of kernel number and kernel size, respectively [32]. As only a few of the N-assimilation genes have been characterized by their biological functions and allelic variations in wheat, their contribution to the genotypic difference in NUE has yet to be explored in the future.

It is worthy to notice that several N transport and assimilation genes are located in the interval of meta-QTL for the yield-related trait under drought and heat stress (Table 2). Wheat crops are grown in environments that are prone to drought and heat stress [33]. Drought and heat stresses are known to have adverse effects on N uptake and assimilation [34,35,36]. For example, short-term heat stress greatly inhibited NR activity and photosynthetic N use efficiency of wheat plants [35]. Drought stress considerably reduced the contents of total protein, Rubisco, and GS2 isoenzyme in wheat leaves, and GS has been suggested as a good indicator in characterizing wheat cultivars in terms of drought stress tolerance [34]. Recently, overexpression of wheat cytosolic and plastid glutamine synthetases has been shown to enhance drought tolerance in tobacco (*Nicotiana tabacum*) [37]. As such, selecting for N use efficient wheat varieties may increase yield stability under drought and heat stress conditions.

## 5. N Uptake and Assimilation for Increasing Spikes and Grain Number

To obtain higher wheat yield and GPC with less N input, it is important to understand the timing of N uptake and wheat development in relation to the formation of yield components. The grain yield can be expressed as:Grain yield = spike number × spikelet number per spike × fertile floret number per spikelet × grain weight

Wheat development can be divided into seedling growth, tillering, stem elongation, booting/ear emergency, flowering, and ripening, the tillering and stem elongation stages are critical in determining fertile spikes and grain number, while the duration and rate of grain filling determine 1000-grain weight [38]. It is well known that N fertilizer increases grain yield mainly by increasing spike number and grain number. N application rate and timing greatly affected spike number, spikelet number per spike, and fertile floret number per spikelet [39,40,41,42], and N concentration and accumulation in both spike and non-spike organs were significantly positively correlated with grain number at stem elongation [40]. As such, N application and efficient N uptake at the relative early growth stage are then critical for increasing spikes and grain number.

It has been reported that winter wheat roots are mostly established during the fall growing season, supporting N uptake demand for rapid above-ground growth in early spring [43], and vigorous early root growth has been shown as a major factor influencing N uptake [44]. Studies on the historical and modern wheat varieties in the wheat main production areas in China revealed that the genetic gain in N uptake (aerial N accumulation amount) mainly raised during stem elongation phase, explaining the significant genetic gains in grain number per spike and per unit area (spike number per unit area × grain number per spike) [45,46]. Therefore, understanding the molecular mechanism underlying the roots’ efficient N acquisition of wheat seedlings may improve NUE and yield by increasing spikes and grain number. Based on our best knowledge, we listed QTL/genes which can increase root growth and nitrate influx rate of wheat seedlings in Table 3, and summarized their effects on N use- and yield-related traits of mature plants grown under field conditions. These QTL/genes mediated root growth, N influx rate, and N assimilation of wheat seedlings, and had positive effects on N use-related traits (N uptake, GNC, N harvest index) and yield-related traits (grain yield, spike number, grain number, 1000-grain weight, and harvest index) of mature plants (Table 3). The major QTL *qTaLRO-2B* coincided with QTL for N uptake and grain yield under low-N conditions, and increased both primary and lateral root length with reduced root diameter, suggesting that this QTL produces a deep and large root system with low carbon cost. In maize, steep, cheap, and deep roots have been suggested as an ideotype for optimizing N acquisition by maize crops [47]. As such, *qTaLRO-2B* may be an ideal locus for wheat breeding with improved NUE and grain yield. The response of root system architecture to N availability is critical for root N foraging [48]. The auxin biosynthetic *Tryptophan Aminotransferase-Related TAR2* is required for low N-stimulated lateral root branching [49,50]. Overexpressing *TaTAR2.1-3A* in wheat enhanced lateral root branching, spike number, grain yield, and N uptake at low- and high-N while reducing *TaTAR2.1* expression through RNAi had the opposite effects [49]. The rice NUE QTL *qDNR1* (*Dull Nitrogen Response 1*) encodes a putative pyridoxal phosphate-dependent transferase and functions in auxin homeostasis, knockout of *OsDNR1* promoted auxin biosynthesis and consequently induced AUXIN RESPONSE FACTOR-mediated activation of N uptake and N-metabolism genes, leading to improved rice yield under moderate levels of N fertilizer input [51]. As such, auxin metabolism and signaling have a crucial role in regulating N use in cereal crops.

Although vigorous early root growth has been shown as a major factor influencing N uptake, the high-vigor wheat lines (38–19, 92–11, and CV97) showed a lower nitrate influx rate than the low-vigor commercial variety Janz, suggesting that the genotypic variation in nitrate uptake capacity (per unit of the root) offset differences in morphological traits and should be considered in efforts to improve N uptake [59]. Vigorous root growth with a high nitrate uptake capacity is then promising in increasing NUE and grain yield. The nitrate transporter *TaNRT2.5-3B*, N-assimilation gene *TaGS2-2A*, and the transcription factors *TaNFYA-6B* and *TaNAC2-5A* have been shown to enhance root growth as well as nitrate influx rate of wheat seedlings, and consequently increased N use- and yield-related traits (Table 3).

It has been mentioned above that overexpressing *TaNADH-GOGAT-3B* increased N uptake, spike number, and grain yield. The expression of *TaNADH-GOGAT* was negatively regulated by the ABRE-binding factor (ABF)-like leucine zipper transcription factor *TabZIP60*-*6D*, reducing *TabZIP60* expression though RNAi increased NADH-GOGAT activity, N uptake, spike number, and grain yield, while overexpressing *TabZIP60*-*6D* had opposite effects [28]. In rice, *OsARE1* is a genetic suppressor of a rice *fd-gogat* mutant defective in N assimilation, and knockout of *ARE1* increased rice yield under low- and high-N conditions [60]. Recently, CRISPR/Cas9-mediated targeted mutagenesis has been used to generate a series of transgene-free mutant lines with partial or triple-null mutation of the three *TaARE1* homoeologs from the elite winter wheat variety Zhengmai 7698. All the mutant lines displayed higher spike number, grain number, 1000-grain weight, and grain yield than the wild type under field conditions [61]. As such, the identification of genes negatively regulating N assimilation provides the opportunity to engineer wheat with improved NUE and yield through a genome editing approach.

## 6. N Uptake and Assimilation for Increasing Yield and Grain Protein Concentration

One of the challenges in wheat breeding is to simultaneously improve both yield and GPC because of the strong negative relationship between these two traits [62]. Several studies on historical and modern wheat varieties revealed significant genetic gain in grain yield, but not in GPC (Table 1). However, there is a genotypic difference in grain protein deviation, the deviation from the regression line between grain yield and GPC, and grain protein deviation has been suggested as a selection criterion for wheat breeding with increased grain yield and GPC [16,62,63,64]. QTL for grain protein deviation has been detected in common wheat and durum wheat, and several N-related genes were found to be linked with QTL for grain protein deviation [65,66], suggesting a genetic opportunity for increasing grain yield and GPC simultaneously.

Grain N originates from remobilization of N accumulated in vegetative tissues pre-anthesis and N uptake post-anthesis. Several studies pointed out the importance of post-anthesis N uptake in determining grain protein deviation and GPC [16,63,64,67,68]. It has been reported that post-anthesis N uptake was associated with expression of *TaNRT2* members, such as *TaNRT2.1* and *TaNRT2.2* on group 6 chromosomes, *TaNRT2.3* on group 2 chromosomes, *TaNRT2.4* on group 1 chromosomes, *TaNRT2.5* on group 3 chromosomes, *TaNRT2.6* on group 7 chromosomes, and *NRT2* partner genes *TaNAR2* [21,62,69]. The TaNRT2.5-TaNAR2.1 complex had nitrate transport activity in *Xenopus* oocytes and was located to the tonoplast in a tobacco leaf transient expression system. *TaNRT2.5* and the root-specific *TaNRT2.1* had comparable mRNA abundance in roots at the grain filling stage. Overexpressing *TaNRT2.5-3B* increased post-anthesis N uptake and grain yield without reducing GNC [21]. The nitrate-inducible NAC transcription factor TaNAC2-5A directly regulates the expression of *TaNRT2.5* and *TaNRT2.1*, and overexpression of *TaNAC2-5A* significantly increased nitrate influx rate, N uptake, GNC, and grain yield under both low- and high-N conditions [58]. As such, manipulating *TaNRT2* expression shows great potential in increasing grain yield and GPC simultaneously.

Although several QTL for root system architecture-related traits were associated with N use-related traits [11], their contribution to N use was largely not validated. A recent study has investigated the effects of *QMrl-7B* on N use and yield by using near-isogenic lines (NILs). *QMrl-7B* was a major stable QTL controlling the maximum root length of wheat seedlings and was linked with a QTL for N uptake at maturity [55]. When a set of *QMrl-7B* NILs with superior and inferior alleles was grown under field conditions in two consecutive growing seasons, the NILs with the superior allele had larger and deeper root system from seedling stage to maturity, greater N uptake after stem elongation, and higher grain yield, 1000-grain weight, harvest index, GNC and N harvest index at maturity than those with the inferior allele under high- and low-N conditions [56]. These results indicate that *QMrl-7B* is valuable in increasing grain yield and GPC. Besides *QMrl-7B*, several key determiners for wheat development also have a crucial role in controlling root system architecture. The Green Revolution *Rht1* genes have been found to reduce root biomass [70]. Whereas introducing the 1RS alien translocation into modern varieties can greatly increase shallow and deep root biomass and uptake of N [71,72]. The *VERNALIZATION1* (*VRN1*) gene responsible for spring-winter growth habit is a key determiner in controlling root system architecture and NUE in wheat [73,74], the winter allele of *VRN1* leads the winter wheat a narrower root angle than the spring wheat [73]. Since the *RHT1* and *VRN1* genes are important for wheat adaptation to local growing conditions, understanding their interactions with N signaling will facilitate the precise design of N-efficient wheat varieties according to the prevalent allele(s) of *RHT1* and *VRN1* genes in different wheat ecological regions.

It has been reported that GS-mediated N assimilation is one of the main checkpoints controlling the N status of the plant and N remobilization for grain filling [75]. In wheat leaves, GS2 is the major GS isoform [76,77,78], but it experiences rapid loss during grain filling [76,78]. Two studies in China showed that wheat breeding increased grain yield but reduced GNC, and increased N uptake during stem elongation phase but not after anthesis [45,46], possibly because that wheat breeding increased leaf GS activity at stem elongation and anthesis but not at grain filling [46]. As such, low GS2 activity during grain filling may be a limiting factor for increasing post-anthesis N uptake and GPC. This assumption was evidenced by the study of transgenic expressing the elite allele *TaGS2-2Ab* under the control of its native promoter. Compared with the wild type, the transgenic lines had higher expression of *TaGS2* in shoots and roots of the seedlings, and higher GS2 protein abundance and GS activity in flag leave when checked at 14 days post-anthesis [27]. The increased grain yield, GNC, harvest index, and N harvest index of the transgenic lines meet the requirement for an N efficient wheat ideotype [79]. These merits of the transgenic lines were associated with increased root growth and nitrate influx rate possibly by up-regulating *TaNRT2.1* and *TaNPF6.3* expression, N uptake before and after anthesis, N remobilization, and leaf functional duration (Figure 1). The expression of *TaGS2-2A* was directly regulated by *TaNAC2-5A*, overexpression of *TaNAC2-5A* increased grain yield, GNC, and N harvest index [58]. As such, the identification of genes regulating *TaGS2* expression and GS2 activity may provide more gene resources for improving grain yield and GPC.

Besides *GS2*, NR encoding genes also showed potential in improving grain yield and GPC. All the six *NR1* genes are located in the interval of the meta-QTL for N use- and yield-related traits (Table 3). Compared with the low NR wheat genotype, the high NR genotype had increased N harvest index and GNC without yield penalty [80]. Overexpression of a tobacco *NR* gene in wheat increased 1000-grain weight and GPC [81]. In rice, overexpression of the elite allele of *OsNR2* from *indica* rice increased grain yield and N concentration in panicle [29].

Efficient N uptake and assimilation increase spikes and grain numbers, and efficient N uptake and remobilization during grain filling benefit higher grain weight and grain protein concentration. Increasing *TaGS2* expression enhanced N uptake before and after anthesis, and allocated more N to grains, showing its crucial role in breeding N efficient wheat ideotype. The nitrate transporter *TaNRT2.5-3B*, and the transcription factors *TaNFYA-6B* and *TaNAC2-5A* enhanced root growth as well as nitrate influx rate of wheat seedlings, and thus increased spike number and grain yield. This working model was derived from the study by [27]. DPA days postanthesis, *Pn* net photosynthesis rate.

## 7. Future Perspectives

The goal of improving NUE is to maximize yield potential and GPC economically and environmentally friendly. To achieve this goal, it is required to understand at the molecular level how N signals affect tiller/spike number, spikelet number per spike, fertile floret per spikelet, and allocation of assimilates to grains. Although a larger number of QTL for N use- and yield-related traits and N-responsive genes have been reported, only a few of them have been molecularly characterized owing to wheat’s huge and complicated genome. Considering the complex gene network governing NUE and yield, and the relatively small number of primary N assimilation genes, identification of genes regulating the expression and activity of primary N assimilation genes will facilitate wheat breeding with improve NUE, yield, and GPC through marker-assisted selection and genome editing approaches.

## Figures and Tables

**Figure 1 plants-11-00492-f001:**
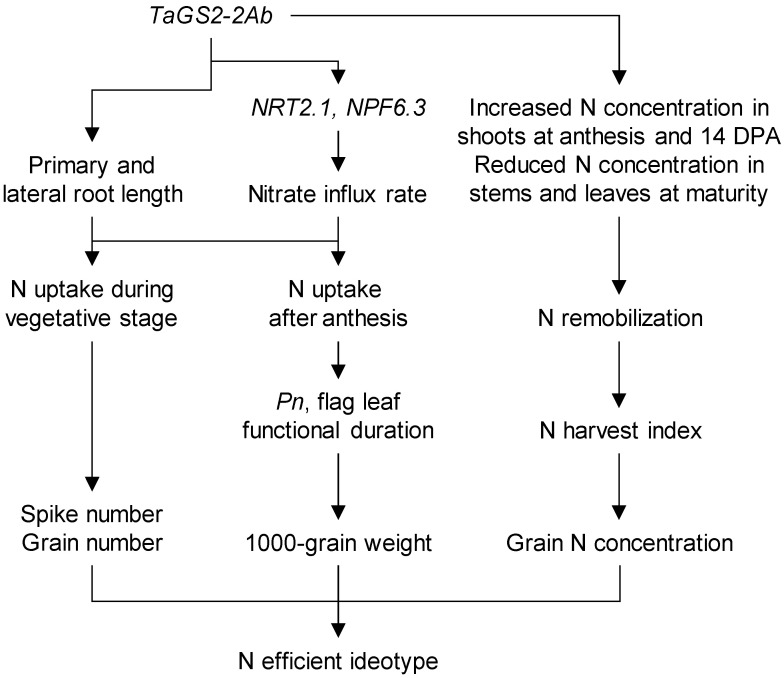
N efficient wheat ideotype mediated by the elite allele *TaGS2-2Ab*.

**Table 1 plants-11-00492-t001:** The rates of genetic gain (% per year) for yield- and N-use-related traits in wheat.

Area	China	European	The United States	South-Eastern Europe
Variety released year	1937~2012	1985~2010	1960~2014	1936~2016
Reference	[14]	[15]	[16]	[13]
Grain yield	0.65%	0.45%	0.331% in 2012 0.761% in 2013	+0.31% at LN +0.34% at HN
Biomass yield	Stable			
Grain number	0.16%			
1000-grain weight	0.38%	Stable		
Spike number	Stable	Stable		
Harvest index	0.62%	0.13%		0.24% at LN 0.28% at HN
Grain N/protein concentration	Stable	Stable		−0.10% at LN −0.10% at HN
N harvest index	0.17%	0.12%	0.05% in 2012 0.20% in 2013	0.07% at LN 0.07% at HN
N uptake efficiency	0.25%	Not evaluated	0.076% in 2012 0.165% in 2013	0.15% at LN 0.12% at HN
N utilization efficiency	0.31%	0.20%	0.115% in 2012 0.367% in 2013	0.15% at LN 0.20% at HN
Post-anthesis N uptake			2.95% in 2012 0.485% in 2013	0.38% at LN 0.51% at HN
N partial factor productivity	0.64%			
N agronomic efficiency	0.68%			

HN and LN indicate high- and low-N treatment, respectively.

**Table 2 plants-11-00492-t002:** The N uptake and assimilation genes are linked with meta-QTL for N use- and yield-related traits.

Gene	PhPosition on RefSeq V1.0 ^a^	Gene Name	Meta-QTL for Yield [23]	Meta-QTL for Yield ^b^ [24]	Meta-QTL for NUE/RSA [11]
N transport					
TraesCS4B02G029600	4B:21928977-21936945	*NPF2.4-4B*	✓		
TraesCS4D02G026800	4D:11862455-11867655	*NPF2.4-4D*	✓		
TraesCS3B02G095900	3B:64314648-64316600	*NPF6.1-3B*	✓		
TraesCS1A02G031300	1A:14519757-14525659	*NPF6.2-1A*			✓
TraesCS1D02G032700	1D:13040668-13043201	*NPF6.2-1D*	✓		
TraesCS2D02G182900	2D:127113821-127117419	*NPF7.1-2D*			✓
TraesCS7D02G297000	7D:374564870-374572229	*NRT1.1A-7D*		✓	
TraesCS1D02G214200	1D:299575952-299578670	*NRT1.1B-1D2*		✓	
TraesCS5A02G537100	5A:693632703-693642408	*NRT1.1C-5A*	✓		
TraesCS2A02G074800	2A:33054150-33056031	*NRT2.3-2A*		✓	✓
TraesCS2D02G073500	2D:30787486-30789242	*NRT2.3-2D*	✓	✓	
TraesCS1D02G035700	1D:16504613-16506169	*NRT2.4-1D*	✓		
TraesCS3A02G254000	3A:475304797-475306341	*NRT2.5-3A*	✓		
TraesCS7A02G428500	7A:621910950-621913739	*NRT2.6-7A*		✓	
TraesCS7B02G328700	7B:583923053-583926829	*NRT2.6-7B*			✓
TraesCS7D02G420900	7D:540617018-540627808	*NRT2.6-7D*	✓		
NRT2 cluster	6A:15727844-16410137	13 *NRT2*s			✓
N assimilation					
TraesCS4A02G376700	4A:651365942-651370213	*NR1.2-4A*		✓	✓
TraesCS7A02G078500	7A:43180494-43185382	*NR1.2-7A*			✓
TraesCS7D02G073700	7D:43212472-43217253	*NR1.2-7D*			✓
TraesCS6A02G017500	6A:8694483-8700180	*NR1.1-6A*	✓		
TraesCS6B02G024900	6B:15128191-15134280	*NR1.1-6B*	✓		
TraesCS6D02G020700	6D:8149929-8155961	*NR1.1-6D*	✓		✓
TraesCS6A02G333900	6A:564882616-564886300	*NiR-6A*	✓	✓	
TraesCS6A02G298100	6A:531394366-531398363	*GS1.1-6A*	✓		
TraesCS4B02G04740	4B:34722272-34725256	*GS1.3-4B*	✓	✓	
TraesCS4D02G047400	4D:22946578-22949866	*GS1.3-4D*	✓		
TraesCS2A02G500400	2A:729293649-729297303	*GS2-2A*	✓		
TraesCS2B02G528300	2B:722629776-722634436	*GS2-2B*	✓	✓	
TraesCS2A02G130600	2A:78328734-78346097	*FD-GOGAT-2A*		✓	
TraesCS2D02G132900	2D:78375985-78392563	*FD-GOGAT-2D*		✓	
TraesCS3A02G266300	3A:490922100-490932750	*NADH-GOGAT-3A*	✓		✓

^a^ The physical positions were retrieved from http://wheat.cau.edu.cn/TGT/, accessed on 10 December 2021. ^b^ The meta-QTL was analyzed based on QTL related to wheat yield under irrigated, drought- and/or heat-stressed conditions. RSA root system architecture.

**Table 3 plants-11-00492-t003:** QTL/genes controlling root system architecture (RSA)- and N uptake-related traits of wheat seedlings.

QTL/Gene Name	RSA-Related Traits of Wheat Seedlings	N Use-Related Traits	Yield-Related Traits	Reference
*qTaLRO-2B*	RDW-HN&LN, MRL-HN&LN, PRL, PRE, LRL, TRL, RD(-)	NUP	GY-LN, BY-LN,	[52,53,54,55]
*qMRL-7B*	RDW, MRL, TRL, and RT at HN and LN	NUP and NHI at HN and LN	GY and TGW at HN and LN	[55,56]
*TaTAR2.1-3A*	LRL and RT at HN and LN	NUP at HN and LN	BY, GY, and SN at HN and LN	[49]
*TaNFYA-6B*	LRL	*TAR2*, *NPF8.3-7D*, *NRT2.1-6B*. Nitrate influx rate at LN, GNC-HN, NUP at HN, and LN	BY, GY, and SN at HN and LN	[57]
*TaNAC2-5A*	RDW at LN Seedling vigor	*NPF7.9-6D* (former named as NPF7.1-6D), *NRT2.1-6B*, *TaNRT2.5, TaGS2-2A*. Nitrate influx rate at HN and LN. GNC, grain nitrate concentration, NUP, and NHI at HN and LN	GY and SN at HN and LN. SDW and TN of wheat seedlings at HN and LN.	[21,58]
*TaNRT2.5-3B*	Seedling vigor, LRL-HN&LN, PRL-LN	Nitrate influx rate at HN and LN. NUP, grain nitrate concentration	GY, SN	[21]
*TaGS2-2Ab*	MRL and LRL at HN and LN	*NRT2.1*, *NPF6.3*, Nitrate influx rate at HN and LN; SNC at stem elongation, flowering, 14 days after flowering at HN and LN; LNC and *Pn* of flag leaves during grain filling at HN and LN; NUP, GNC, NHI, and LNC (-) at maturity at HN and LN.	GY, SN, GN, TGW, and HI at HN; net photosynthesis rate of flag leaves at HN and LN	[27]
*TaNADH-GOGAT-3B*	MRL, LRL	NUP	BY, GY, SN	[28]

RSA-related traits: MRL maximum root length, PRL primary root length, LRL lateral root length, TRL total root length, PRE primary root elongation rate, RD, root diameter, RT root tip number. N use-related traits: NUP N uptake, NCT, N concentration, GNC grain N concentration, SNC shoot N concentration, LFC leaf N concentration, *Pn* net photosynthesis rate, NHI N harvest index. Yield-related traits: BY biomass yield, GY grain yield, SDW shoot dry weight, SN spike number, TN tiller number, TGW 1000-grain weight, HI harvest index. N treatment: HN high N treatment, LN low N treatment. The negative mark in paragraph (-) after a trait indicates the negative effect of the QTL/gene on that trait.

## Data Availability

All data cited in the study are public available.

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
