# Peer review of "Genetic Control of Efficient Nitrogen Use for High Yield and Grain Protein Concentration in Wheat: A Review"

_plants, 2022, doi:10.3390/plants11040492_

Round 1

Reviewer 1 Report

The work is really valuable and relevant and there is no doubt about the publication.

But there is also a downside - the text is overloaded with abbreviations, which makes the text difficult and inconvenient to read. In addition, there are sometimes unexplained abbreviations (they are marked in the text, as well as some minor errors or suggested insertions).

I would also suggest a brief overview of the effects of meteorological extremes on plant phases for N uptace. Is  the genome  coud control / mitigate meteo-extremes ?...

Author Response

The work is really valuable and relevant and there is no doubt about the publication.

Thank you very much for your comments and revision suggestion. I have revised the manuscript according to your comments in the attached PDF.

But there is also a downside - the text is overloaded with abbreviations, which makes the text difficult and inconvenient to read. In addition, there are sometimes unexplained abbreviations (they are marked in the text, as well as some minor errors or suggested insertions).

I deleted the abbreviations of yield-related traits, but remained the abbreviations of N use-related traits.

I checked the manuscript and now there was no unexplained abbreviation.

I would also suggest a brief overview of the effects of meteorological extremes on plant phases for N uptake. Is the genome could control / mitigate meteo-extremes ?.

This is a very important point. I briefly introduced the relation of N use with drought and heat stress as shown below. I hope that this brief overview addressed this comment.

It is worthy to notice that a number of N transporter and assimilation genes located to the intervals of meta-QTL for yield-related train under drought stress and heat stress (Table 2). Wheat crops are grown in environments that are prone to heat and drought stress [33]. Drought and heat stresses are known to have adverse effects on N uptake and assimilation [34-36]. For example, short-term heat stress greatly inhibited NR activity and photosynthetic N use efficiency of wheat plants [35]. Drought stress considerably reduced the contents of total protein, Rubisco and GS2 isoenzyme in wheat leaves, and GS has been suggested a good indicator in characterizing wheat cultivars in terms of drought stress tolerance [34]. Recently, overexpression of wheat cytosolic and plastid glutamine synthetases has been shown to enhanced drought tolerance in tobacco [37]. As such, selecting N use efficient wheat varieties may increase yield stability under drought and heat stress conditions.

Reviewer 2 Report

Interesting work reviwewing some aspects of the N utilisation efficiency from a molecular biological point of view.

Minor comment: There is a small typo in page 10 line 8 (double comma).

Author Response

Thank you for your comments. I have revised the typo in page 10.

Reviewer 3 Report

I reviewed the article titled: " Genetic control of efficient nitrogen use for high yield and grain protein concentration in wheat: a review " and I found it riveting and well-prepared. Authors  have demonstrated extensive comprehension of environmental key-issues in the field of agriculture and plant genetics. The article is a well needed source of condensed agri-related knowledge. Information collected by this research group can be used in far-reaching nitrogen management and plant genetics, which I find scientifically significant. Based on my experience and according to the actual state of knowledge I accept this article after minor revision (mainly on introductive part of the manuscript). Please check and verify the following points:

 [22-23] The authors should have indicated N role in plant nutrition before introducing fertilizer role essentiality. Please do correct that. Without that it doesn't seem right for me. It would be most aptly to add some information about wheat nutritional requirements thus smoothly indicating this essential role of N fertilizing that authors have mentioned.

 [24-25] “(..)and was lost to the environment.” I strongly suggest to add more specific information about N loses, ex. NO3-,NH4+ leaching, NH3 emissions etc..

[26-28]  “As such, it is required to increase wheat yield in economically and environmentally friendly manner, by breeding wheat with improved N use efficiency (NUE) and designing variety-specific recommendations of N fertilizer management.”   This fragment should be changed because those are not the only  applicable ways of increasing wheat yield in general.  It’s better to state it this way: Therefore, it is required to increase wheat yield in economically and environmentally friendly manner. Such as breeding wheat with improved N use efficiency (NUE) and designing variety-specific recommendations of N fertilizer management.

[38] “.. Nt is total N in the plant at maturity per unit area (grain + stove)”  should be changed into “.. Nt is total N in the plant (grain + stove) at maturity per unit area” 

[116-117] Please do paraphrase this fragment or delete it because now it seems like it is just repetition from line 24.

All of mentioned flaws are considered minor. Thus, I am positive that if authors apply indicated corrections this paper is worth publishing.

Author Response

Thank you very much for your comments and revision suggestion.

 [22-23] The authors should have indicated N role in plant nutrition before introducing fertilizer role essentiality. Please do correct that. Without that it doesn't seem right for me. It would be most aptly to add some information about wheat nutritional requirements thus smoothly indicating this essential role of N fertilizing that authors have mentioned.

According this comment, I added the following sentences:

Nitrogen (N) is an essential nutrient, and often the most critical yield-limiting factor in crop production [3]. Among the crops, wheat received the highest amount of N fertilizers, with 18.2% of global use, followed by maize with 17.8% and rice with 15.2% [4].

 [24-25] “(..)and was lost to the environment.” I strongly suggest to add more specific information about N loses, ex. NO3-,NH4+ leaching, NH3 emissions etc..

 According this comment, I added the following sentences:

Fertilizer N not recovered by crops can cause environmental problems of nitrate pollution of waters and the pollution of the atmosphere with nitrous oxide, other oxides of N, and ammonia [6].

[26-28]  “As such, it is required to increase wheat yield in economically and environmentally friendly manner, by breeding wheat with improved N use efficiency (NUE) and designing variety-specific recommendations of N fertilizer management.”   This fragment should be changed because those are not the only  applicable ways of increasing wheat yield in general.  It’s better to state it this way: Therefore, it is required to increase wheat yield in economically and environmentally friendly manner. Such as breeding wheat with improved N use efficiency (NUE) and designing variety-specific recommendations of N fertilizer management.

Revised as suggested.

[38] “.. Nt is total N in the plant at maturity per unit area (grain + stove)”  should be changed into “.. Nt is total N in the plant (grain + stove) at maturity per unit area” 

Revised as suggested.

[116-117] Please do paraphrase this fragment or delete it because now it seems like it is just repetition from line 24.

 This sentence is changed to:

To obtain higher wheat yield and GPC with less N input, it is important to understand the timing of N uptake and wheat development in relation with the formation of yield components.

All of mentioned flaws are considered minor. Thus, I am positive that if authors apply indicated corrections this paper is worth publishing.